# Complete Solution-Processed Semitransparent and Flexible Organic Solar Cells: A Success of Polyimide/Ag-Nanowires- and PH1000-Based Electrodes with Plasmonic Enhanced Light Absorption

**DOI:** 10.3390/nano12223987

**Published:** 2022-11-12

**Authors:** Jing Wang, Xiangfei Liang, Jianing Xie, Xiaolong Yin, Jinhao Chen, Tianfu Gu, Yueqi Mo, Jianqing Zhao, Shumei Liu, Donghong Yu, Jibin Zhang, Lintao Hou

**Affiliations:** 1Guangdong-Hong Kong-Macao Joint Laboratory for Intelligent Micro-Nano Optoelectronic Technology, School of Physics and Optoelectronic Engineering, Foshan University, Foshan 528000, China; 2Guangzhou Key Laboratory of Vacuum Coating Technologies and New Energy Materials, Siyuan Laboratory, Physics Department, Jinan University, Guangzhou 510632, China; 3State Key Laboratory of Luminescent Materials and Devices, College of Materials Science and Engineering, South China University of Technology, Guangzhou 510640, China; 4Department of Chemistry and Bioscience, Aalborg University, Fredrik Bajers Vej 7H, DK-9220 Aalborg East, Denmark; 5Sino-Danish Center for Education and Research, DK-8000 Aarhus, Denmark; 6Key Laboratory of Materials Physics of Ministry of Education, School of Physics and Microelectronics, Zhengzhou University, Zhengzhou 450052, China

**Keywords:** polyimide, Ag nanowires, plasmonic, semitransparent, flexible, organic solar cells

## Abstract

Organic solar cells (OSCs) have been widely studied due to the advantages of easy fabrication, low cost, light weight, good flexibility and sufficient transparency. In this work, flexible and semitransparent OSCs were successfully fabricated with the adoption of both polyimide/silver nanowires (PI/AgNW) and a conducting polymer PEDOT:PSS named PH1000 as the transparent conductive electrodes (TCEs). It is demonstrated that PI/AgNW is more suitable as a cathode rather than an anode in the viewpoint of its work function, photovoltaic performance, and simulations of optical properties. It is also found that the light incidence from PH1000 TCE can produce more plasmonic-enhanced photon absorption than the PI/AgNW electrode does, resulting in more high power conversion efficiency. Moreover, a high light transmittance of 33.8% and a decent efficiency of 3.88% are achieved for the whole all-flexible semitransparent device with only 9% decrease of resistance in PI/AgNW after 3000 bending cycles. This work illustrates that PI/AgNW has great potential and bright prospect in large-area OSC applications in the future.

## 1. Introduction

Photovoltaic (PV) technology plays an important role in achieving solar energy conservation and decreasing fossil fuels consumption due to its clean and sustainable-development advantages [1,2,3]. Compared to the traditional inorganic solar energy materials with poor flexibility [4,5,6], polymeric semiconducting materials possess the super mechanical flexibility due to the entanglement and relaxation of such long-chain molecules with conjugated backbones equipped with solubilizing side-chains [7,8,9]. For constructing the flexible organic solar cells (OSCs), transparent conductive electrodes (TCEs) with good flexibility are equally essential. The most commonly used TCEs include carbon nanotube [10], ultra-thin metal [11], conductive polymer [12,13], and metal nanowires [14,15], among which metal nanowire had been considered as the most potentially flexible one due to their high optical transparency, high conductivity, and low cost [16]. Although silver nanowire (AgNW) has been used as the cathode for flexible thin-film solar cells with many improvements having been made to improve device performance [17,18,19,20,21,22], the many defects such as rough surface and poor adhesion prevent their more widespread use. The reasons for AgNW being a better TCE in semi-transparent OSCs are still in need of further study especially in the case of the semi-embedding polyimide (PI)/AgNW electrode with ultra-lightweight and high thermal/mechanical stability. 

Since scattering and plasma effects produced by AgNW could enhance the optical absorption of OSCs, optical analysis should be used for guiding the optimization of each layer in device. The enhancement effect of light absorption can be simulated by the finite-difference time-domain (FDTD) method [23,24,25], which has been used in PET/AgNW electrode [26], ZnONPs/AgNWs/ZnONPs electrode [27] and AgNW electrodes in OSCs [28,29,30]. This method is still an important tool towards accurately predicting the optical absorption performance in PI/AgNW based OSCs. In addition, the reasonable molecular level interactions between donor and acceptor is necessary to achieve high-efficiency OSCs, as verified by multiscale characterization techniques and photovoltaic device physics [31,32,33].

In order to further study the PI/AgNW based flexible and semitransparent OSCs with the validation of the proposed design, simulations and experiments were carried out for a typical active system of poly(4,8-bis[5-(2-ethylhexyl)thiophen-2-yl]benzo[1,2-b:4,5-b′]dithiophene-2,6-diyl-alt-3-fluoro-2-[(2-ethylhexyl)carbonyl]thieno[3,4-b]thiophene-4,6-diyl) (PTB7-Th) donor and [6,6]-phenyl-C71-butyric acid methyl ester (PC_71_BM) acceptor in this work. Since the high roughness of AgNW would easily cause short-circuit to the devices, we adopted a simple method of embedding AgNW into PI for reducing the roughness. Such PI/AgNW conductive film has the same high optical transmittance and low resistance as the most widely used stiff ITO electrode. To determine the best device structure with PI/AgNW electrode and to utilize the plasma effect of AgNW on optical absorption enhancement, both normal and inverted opaque flexible OSCs were studied by optical- and electrical-analysis in depth. The surface plasmon excited by light irradiation on AgNW can improve the optical absorption of the active layer compared with that using the ITO electrode, and the PI/AgNW as the cathode can help the active layer absorb significantly more light than the PI/AgNW as the anode. We then applied the PI/AgNW cathode to building the semitransparent flexible device. The optical analysis of photocurrents generated by light incident from both sides of the device show that semitransparent flexible devices with PI/AgNW cathodes can absorb more light when light is irradiated from the PH1000 side. Such results promise future application of flexible substrate to the production of flexible semitransparent devices.

## 2. Materials and Methods

The active layer consists of PTB7-Th (1-materials, Dorval of Canada) and PC_71_BM (Solenne, Halland of Sweden). AgNW suspension (1 wt%) in ethanol was purchased from Zhejiang Kechuang Advanced Materials Co. Ltd. The fabrication of PI/AgNW TCE is depicted in Figure 1a. A dispersion of AgNWs (20–30 nm) in ethanol was spread onto a plasma-treated glass substrate and the wet solution was vaporized in the air for 3 minutes. Then the AgNW coated glass substrate was washed with deionized water to remove the solvent residue and baked at 100 °C for 5 min. Then PI in dimethylformamide (DMAc) solution (14 wt%) was spin-coated on the AgNW film under 700 rpm for 9 s, and immediately transferred into a vacuum drying oven and cured according to one-step or multi-step heating processes. Finally, the edge of PI/AgNW film was scraped off and soaked in deionized water for several minutes, leading to smooth PI/AgNW TCE after peeling off.

Flexible OSCs were prepared by spin-coating with PI/AgNW as the TCE. Zinc oxide nanoparticles (ZnO NP) (Avantama, Zurich, Switzerland) or HILE-100 (Clevios, Hanau, Germany) were spin-coated on top of PI/AgNW as the electron transport layer (ETL) or the hole transport layer (HTL) and annealed at 80 °C for 2 min. The active PTB7-Th:PC_71_BM blend with a weight ratio of 1:1.5 was dissolved in a mixed solvent of chlorobenzene (CB): 1,8-diiodooctane (DIO) (97:3 by volume) for about 12 h at 60 °C and then spun-coated onto the substrates in a glove box. Next, MoO_3_ and Ag were thermally deposited under vacuum less than 4 × 10^−4^ Pa. The active area was defined as a pixel size of 0.14 cm^2^. The flexible semi-transparent device can be obtained by replacing Ag metal with 100 nm-thick PH1000 (Clevios) with adding 0.5% surfactant additive (FS-30) and 5% DMSO and annealing at 60 °C for 60 s.

The sheet resistance of AgNW/PI TCE was measured by a four-point probe system. The performance of OSCs was characterized by a Keithley 2400 source meter under the illumination of AM1.5G solar simulators with an intensity of 100 mW/cm^2^ (Sun 2000, Abet, Milwaukee, Wisconsin of America). The light intensity of the solar simulator was calibrated using a standard silicon photodiode (NIMMS1101, National Institute of Metrology, Beijing, China), which was also used as the reference solar cell in the EQE measurement to obtain a transformation coefficient that is compared to the current-voltage (*J*–*V*) curve. A quartz crystal thickness/ratio monitor (model STM-100/MF, Sycon, Hilscher, Hattersheim of Germany) was used to measure the thickness of films. Transient photovoltage (TPV) was performed by the Paios instrument. A FDTD method has been used for solving Maxwell’s equations via the commercial FDTD Solution software from Lumerical [34,35,36].

In optical modeling, since both the device structure and the light source are axisymmetric, symmetric/antisymmetric boundary conditions were selected, with X-axis set as the antisymmetric boundary and Y-axis as the symmetric one. The main purpose was to reduce the simulated area by using symmetry, so that the actual calculation area is reduced to 1/4 of the simulated one in order to reduce the memory requirement. To avoid re-introducing reflections, the boundary conditions of perfectly matched layers (PMLs) are used along the z-direction to absorb all light waves moving toward the outside of the simulation area. For the homogeneous active layer, a mesh size of 5 nm was used. The incident light is a plane wave with a wavelength range from 250 to 1000 nm, which encompasses the solar spectrum. The optical constants of AgNW, ITO, ZnO and PTB7-Th:PC_71_BM are taken from palik’book and refractiveindex database [37].

## 3. Results and Discussion

Colorless and soluble PI [38] was used as a flexible substrate of TCEs due to its excellent mechanical properties, high thermal stability, and high transmittance. In this study, AgNW/AgNW TCEs were made using a recently reported embedding procedure [39], in which AgNW was drop-casted on a glass substrate followed by PI solution spin coating and curing. For the sake of studying the effect of the curing process on the conductivity of the resulting TCEs, the temperature-time curves in the one- and multi-step heating processes and the relative resistance change rate vs bending cycle curves of the corresponding PI/AgNW TCEs are shown in Figure 1b,c. It can be seen that the PI heating process has a big effect on the conductivity stability of TCEs in many bending cycles. Generally, the resistance increases with the increasing bending cycles, which can be attributed to the deterioration of the conductive network during bending. The PI/AgNW TCE via multi-step heating process shows higher conductivity stability than that via the one-step heating process, indicating that the AgNW conductive network in the PI matrix processed by multi-step heating maintains better electric contact than that via the one-step heating process. The resistance of AgNW/PI TCE processed by multi-step heating increases only 9% after 3000 bending cycles comparing to 33% increase for the one via one-step heating process. Furthermore, the high conductivity of 3560 S·cm^−1^ for the AgNW/PI TCE is comparable with the commerical ITO TCE with inherent disadvantages of high cost and poor bendability. Moreover, the very low roughness (RMS = 1.210 nm) of AgNW/PI TCE in height is achieved with evenly distributed AgNW on the PI surface in a large area of 20 × 20 μm^2^, which is very favorable for solar device fabrication by avoiding cell breakdown and leakage (Figure 1d). Besides, the surface energies of AgNW/PI TCEs with and without O_2_ plasma treatment were measured based on Owens equation [40], in which the surface energy of the plasma-treated TCE (67.1 mN m^−1^) is much higher than that of the non-plasma-treated one (24.9 mN m^−1^), indicating the quality of the subsequent anode or cathode interfacial layer by coating can be effectively controlled using this method.

The prepared PI/AgNW TCE can be used as either anode or cathode in flexible OSCs. However, the facts that PI/AgNW is more suitable as the anode or as the cathode are in dispute. For sake of understanding this problem, two types of PI/AgNW based OSCs with the normal structure (S1: PI/AgNW/HILE-100/PTB7-Th:PC_71_BM/ZnONP/Ag) and the inverted structure (S2: PI/AgNW/ZnONP/PTB7-Th:PC71BM/MoO_3_/Ag) are designed, comparing to the corresponding ITO-based OSCs with the normal structure (S3: Glass/ITO/HILE-100/PTB7-Th:PC_71_BM/ZnONP/Ag) and the inverted structure (S4: Glass/ITO/ZnONP/PTB7-Th:PC_71_BM/MoO_3_/Ag). The chemical structures of PTB7-Th and PC71BM are shown in Figure 1e, which are very successful optically active materials.

The optical simulation of absorbed energy distribution was carried out for above four types of devices. It was found that a significant amount of energy energy can be absorbed in the center of the photoactive layer for the S2 device, whereas the majority of the absorbed energy is distributed at the edge of the active layer when PI/AgNW acts as the anode for the S1 device (Figure 2a,b). As we know, photons absorption centralized in the active layer can promote more excitons generation, which is very advantageous to producing more photocurrent. Furthmore, it is apparent that the full-wavelength absorbed power for ITO-based normal and inverted devices is less than that of the corresponding PI/AgNW devices, due to the strong plasma-enhanced light absorption of AgNW. The optical analysis shows that PI/AgNW as the cathode is advantageous to enhancing the optical absorption of the active layer.

As energy level matching is another important factor affecting device performance, we tested the work function of AgNW and the energy levels of ZnONP. As shown in Figure 3a, based on the threshold value of Ecutoff which corresponds to the cut-off edge of secondary electron emission in UPS, the work functions (*Φ*) of ZnONP and AgNW films can be calculated via the formula of *Φ* = *h*_v_ − *E*_cutoff_, where *h*_v_ is 21.22 eV for a He I photon source. Therefore, the work function of AgNW is determined as 4.17 eV. Moreover, as shown in Figure 3b, the energy difference between the Fermi level (E_f_) of ZnONP and the value of the valence band maximum (VBM) is 2.55 eV by linear extrapolation, determining the VBM of ZnONP to 6.8 eV, which is close to the value reported in the literature [41]. The deep VBM of ZnONP can effectively prevent holes from entering the cathode, which is beneficial to reduce carrier recombination at the AgNW cathode. To further calculate the conduction band minimum (CBM) of ZnONP, the absorption spectrum of ZnONP was measured (Figure 3c), determining the CBM of ZnONP to be 3.62 eV. As shown in Figure 3d, the electron injection barrier (0.55 eV) between AgNW and ZnONP is much smaller than that (1.03 eV) between AgNW and HIL E-100, indicating PI/AgNW is more suitable to be a cathode TCE in flexible OSCs [42].

To testify the above points experimentally, we fabricated two types of AgNW-based normal and inverted flexible OSCs in comparison with the ITO-based rigid ones. Figure 3e presents the *J*–*V* curves of S1, S2, S3 and S4 devices measured under simulated AM1.5G illumination. The deduced photovoltaic parameters of all devices including the short-circuit current (J_SC_), open-circuit voltage (V_OC_), fill factor (FF), and power conversion efficiency (PCE) are summarized in Table 1. The PCE of S2 device is nearly one time higher than that of S1 device, indicating the PI/AgNW is more suitable as the transparent flexible cathode rather than an anode in the device. The comparable device performance of PI/AgNW based devices to that of the rigid ITO-based OSCs illustrates that the flexible PI/AgNW TCE has excellent photoelectric characteristics, which is comparable to ITO transparent electrode.

As mentioned above, we found that PI/AgNW is more suitable as cathode TCE according to optical simulation, interface analysis and device performance. To further develop the application of PI/AgNW TCE in solar energy, we designed and fabricated the all-flexible semitransparent solar cell by replacing the opaque Ag anode with PEDOT:PSS (PH1000). The concrete device structure of all-flexible semitransparent solar cell is PI/AgNW/ZnONP/PTB7-Th:PC_71_BM/PH1000 (S5), as depicted in the inset of Figure 4a. To explore the influence of incident light direction on the photocurrent in device, the *J*–*V* curves of all-flexible semitransparent device illuminated from either PH1000 side or PI/AgNW side were measured, as shown in Figure 4a. When the light is irradiated from the PI/AgNW side, a J_SC_ of 8.30 mA/cm^2^ and a PCE of 3.23% are obtained, respectively, which are much lower than 10.21 mA/cm^2^ and 3.88% when the light is irradiated from the PH1000 side. Figure 4b shows the transmission spectra of the S5 device when the light is incident from either PH1000 side or PI/AgNW side. It can be seen that the transmittance intensity of the device is almost the same under different light incident directions with an average visible transmittance as high as 33.8% (inset of Figure 4b). Thus we concluded that the dependency of light absorption on spatial incident light-direction should exist in these devices, which may come from the different extent of plasma absorption enhancement induced by AgNW in the active layer [43]. The simulated absorption curves under different solar radiation incidence further verified above conclusion that the device illuminated from PH1000 side shows stronger absorption (Figure 4c). Moreover, as shown in Figure 4d, the number of extracted charges for the device irradiated from the PH1000 side under different light intensity is much higher than that from the PI/AgNW side. For example, the number of extracted charges in dark (n_0_) is 2.08 × 10^16^ cm^−3^, while the number of extracted charges (n) under the illumination of visible light (70.0 mW/cm^2^) from the PH1000 side and the PI/AgNW side are 3.31 × 10^16^ and 3.10 × 10^16^ cm^−3^ respectively. A high ratio of 0.59 to the number of extracted charges in the darkness (n-n_0_) is obtained when the light illuminates from the PH1000 side in comparison with that of 0.49 when the light illuminates from the PI/AgNW side, illustrating that more charges can be effectively extracted from the active layer when the all-flexible semitransparent device is illuminated from the PH1000 side.

The electric field and full-wavelength absorbed power spectra of the all-flexible semitransparent device S5 from top and bottom illumination are presented in Figure 5. When the light irradiates from PH1000 to AgNW, the plasmon resonance is generated on the AgNW surface, and the electric field of the middle and bottom of the active layer is significantly enhanced (Figure 5a). In contrast, the electric field of the active layer turns weaker when the light irradiates from PI/AgNW to PH1000 since the plasmon resonance at the AgNW surface primarily acts on the PI layer (Figure 5b). It can also be observed that excitons are efficiently generated in the active layer of all-flexible semitransparent solar cell when the light irradiates from PH1000 to AgNW due to absorption enhancement by advantageous nearfield localized surface plasmon resonance (Figure 5c,d), which surely results in the enhancement of photocurrent (Figure 4a). Assuming that there is no carrier recombination, all photogenerated charges should be collected by the electrode. The simulated short circuit current density of the device illuminated from the PH1000 side is 14.53 mA/cm^2^, while 13.84 mA/cm^2^ for the device illuminated from the PI/AgNW side is obtained. The theoretical results are in good agreement with the experimental ones, favorably proving the correctness of our analysis.

## 4. Conclusions

In this study, the PI/AgNW electrode with excellent mechanical and optoelectrical properties was successfully fabricated as a promising alternative to ITO. It was demonstrated that the PI/AgNW electrode is more suitable as the flexible transparent cathode than that of the one as an anode in OSCs, which is investigated deeply by optical simulation, electrical parameter measurement and performance of devices with normal and inverted structures. The modeling results agreed well with the experimental ones, indicating excellent plasmonic properties of AgNW should be exploited more efficiently for surface-enhanced absorption. The higher device efficiency of all-flexible semitransparent OSC when illuminated from the PH1000 anode side than that from the AgNW cathode side (3.88% vs. 3.23%) further confirms the above significant finding. This work demonstrates that PI/AgNW is a promising and desirable TCE in the realization of high performance, low-cost, and large-scale flexible OSCs.

## Figures and Tables

**Figure 1 nanomaterials-12-03987-f001:**
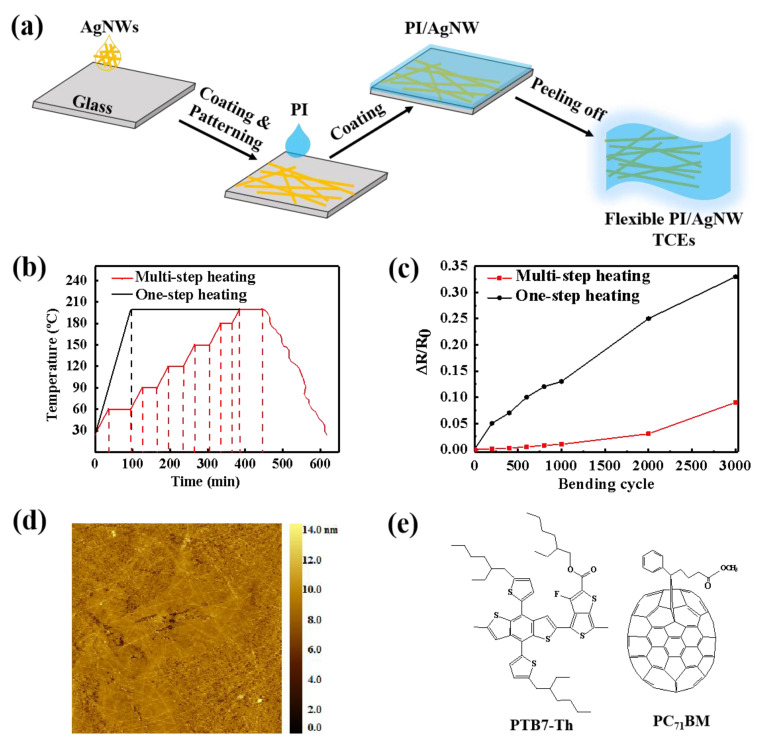
(**a**) Schematic diagram of preparation process of PI/AgNW TCEs. (**b**) Temperature-time curve of PI annealing process. (**c**) Plot of relative resistance change rate vs. bending cycle of PI/AgNW TCEs (bending radius: 2.5 mm). (**d**) AFM image of the PI/AgNW TCE. (**e**) Molecular structures of PTB7-Th donor and PC_71_BM acceptor.

**Figure 2 nanomaterials-12-03987-f002:**
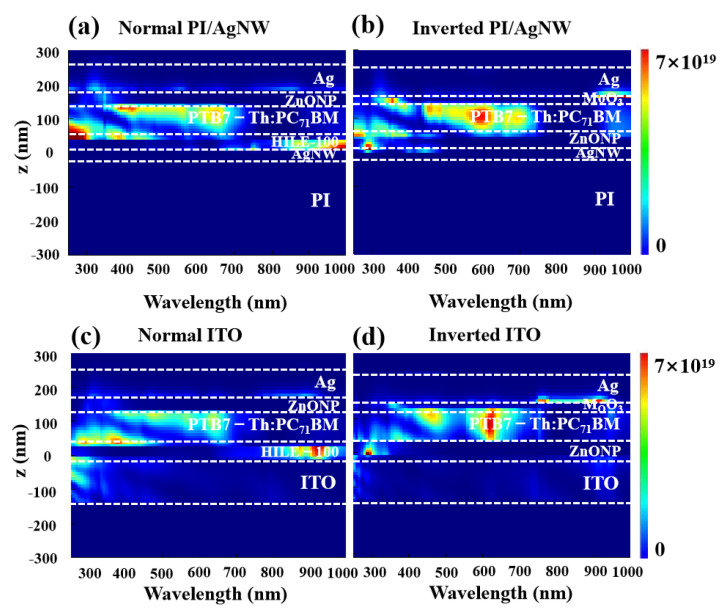
Full-wavelength absorbed power spectra of (**a**) normal (S1), (**b**) inverted (S2) PI/AgNW based devices compared with (**c**) normal (S3) and (**d**) inverted (S4) ITO based devices.

**Figure 3 nanomaterials-12-03987-f003:**
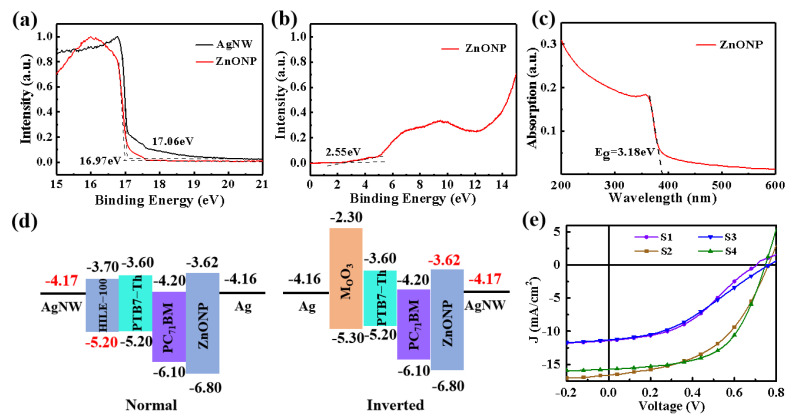
(**a**) UPS secondary electrons cut-off edge spectra of AgNW and ZnONP. (**b**) UPS valence-band maximum spectrum of ZnONP. (**c**) The absorption spectrum of ZnONP. (**d**) Energy band alignment diagrams of devices with PI/AgNW as anode or cathode. (**e**) *J*–*V* curves of flexible S1 and S2 devices in comparison with the control rigid S3 and S4 devices.

**Figure 4 nanomaterials-12-03987-f004:**
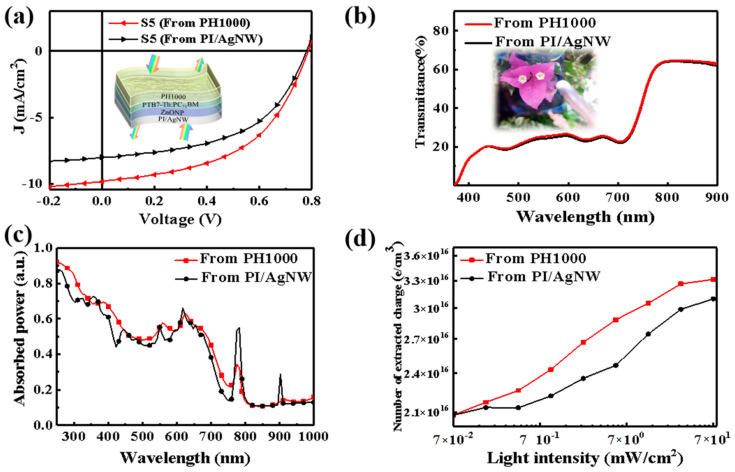
(**a**) *J*–*V* curves of the S5 device illuminated from the PH1000 and the PI/AgNW, respectively. The inset in the figure is the schematic diagram of the structure of semitransparent flexible OSCs (S5) with PI/AgNW TCEs as the cathode. (**b**) Transmittance curve. The inset in the figure is an actual photo of the all-flexible semitransparent device S5. (**c**) Calculated absorbed power curve as well as (**d**) extracted charges vs light intensity curve for the all-flexible semitransparent device illuminated from the PH1000 and the PI/AgNW, respectively.

**Figure 5 nanomaterials-12-03987-f005:**
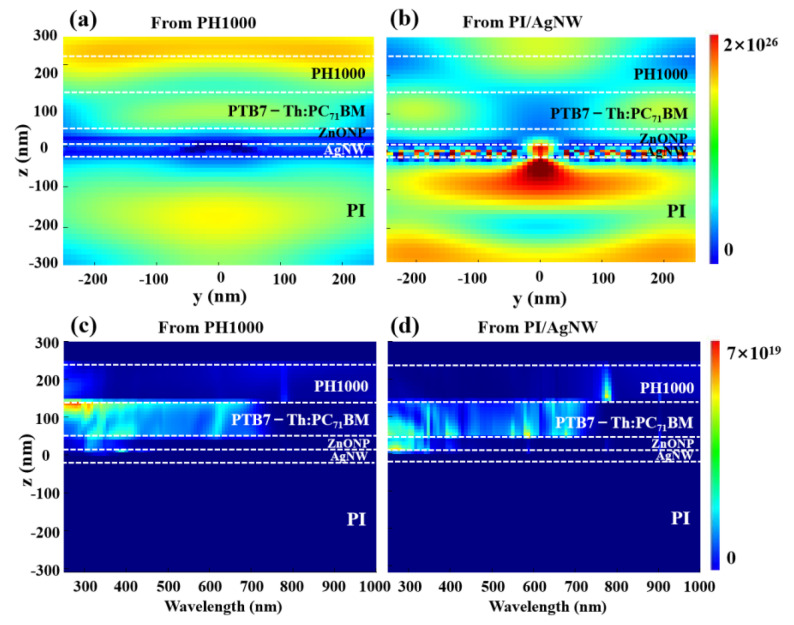
The contour plot of electric field (573 nm) of all-flexible semitransparent inverted solar cell illuminated from (**a**) PH1000 and (**b**) PI/AgNW. Full-wavelength absorbed power spectra of all-flexible semitransparent inverted solar cell illuminated from (**c**) PH1000 and (**d**) PI/AgNW.

**Table 1 nanomaterials-12-03987-t001:** Statistical V_OC_, J_SC_, FF, and PCE of flexible and semitransparent devices under the illumination of AM1.5G (100 mW/cm^2^).

Device	Optical Transparency	Incidence	V_OC_ [V]	J_SC_ [mA/cm^2^]	FF [%]	PCE [%]
S1	No	PI/AgNW	0.71 ± 0.02	11.40 ± 0.09	40.90 ± 0.06	3.32 ± 0.06
S2	No	PI/AgNW	0.76 ± 0.01	16.63 ± 0.05	48.80 ± 0.08	6.17 ± 0.04
S3	No	ITO	0.77 ± 0.02	11.29 ± 0.05	36.40 ± 0.09	3.14 ± 0.05
S4	No	ITO	0.75 ± 0.01	15.74 ± 0.08	57.30 ± 0.09	6.76 ± 0.06
S5	Yes	PH1000	0.79 ± 0.01	10.21 ± 0.07	48.40 ± 0.05	3.88 ± 0.05
		PI/AgNW	0.78 ± 0.02	8.30 ± 0.09	49.96 ± 0.08	3.23 ± 0.07

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
