# Peer review of "Complete Solution-Processed Semitransparent and Flexible Organic Solar Cells: A Success of Polyimide/Ag-Nanowires- and PH1000-Based Electrodes with Plasmonic Enhanced Light Absorption"

_nanomaterials, 2022, doi:10.3390/nano12223987_

Round 1
Reviewer 1 Report
In this manuscript, the authors studied flexible and semitransparent organic solar cells with polyimide/silver nanowires (PI/AgNW) and a conducting polymer PEDOT:PSS (PH1000) as transparent conductive electrodes (TCEs). The authors illustrate that the PI/AgNW is more suitable as a cathode rather than an anode, and found that the light incidence from PH1000 TCE can produce more plasmonic-enhanced photon absorption, loeading to enhanced power conversion efficiency. Up to 33.8% transmittance has been achieved for the OSC device, for which 9% decrease of resistance in PI/AgNW after 3000 bending cycles is obtained. The PI/AgNW electrode provide excellent mechanical and optoelectrical properties, which can be used as an alternative to ITO. Optical absorption, UPS and J-V curves analysis is carried out for these materials. While the manuscript is of potential interest to the material chemists and device physicists in the area of molecular electronics, the morphological characterization of thin films is not provided in the manuscript. Therefore, I suggest the authors to include these details in the introduction, results and discussion.
Comments:
1. Please provide morphological analysis of the PI/AgNW cathode layer, PTB7-Th and PC71BM.
2. Please state the PCE values of best OSC devices in the abstract and in the conclusions.
3. Structural insights and morphological analysis are of direct interest to the synthesis chemists in order to understand the key intermolecular interactions at the interfaces between the photoactive layers and the electrodes (as well within the electrode materials and photoactive materials). There is a recent literature in the NFA organic solar cells (see doi.org/10.1038/s41578-020-00232-5, for PTB7-Th see 10.1002/adfm.201901109,and for high performing PM6:Y6 solar cells, see doi/10.1002/adma.202105943 and doi.org/10.1039/D0EE01896A) about resolving atomic scales interactions and establishing structure property relationships by multiscale characterization techniques and device physics. In particular, wide angle X-ray scattering and solid-state NMR spectroscopy analysis is particularly striking to understand the molecular level interactions in OSC thin films. The authors are encouraged to mention these results and techniques in the introduction.
Author Response
We thank the reviewer for the positive feedback of our work and the constructive suggestions/comments to further improve the quality of our manuscript. We provide the response to each comment below:

Reviewer 2 Report
The authors present an interesting experimental study of fabrication of flexible and semitransparent organic solar cells using polyimide/silver nanowires and a conducting polymer PEDOT:PSS as transparent conductive electrodes.
This paper is written in a good way pointing out very critical things. However, I suggest that the author could modify or add a few things below.
1. In the energy band alignment diagrams, how does the VBM of HILE-100 estimated? Is this the value of VBM from the literature or did the author perform UPS measurements as in the case of ZnONP?
2. Page 5, line 181: Please change the “…Figure 4b,…” to “…Figure 3b,…”.
3. The authors report that the roughness of AgNWs improved when they embedded AgNWs into polyimide. Have the authors performed atomic force microscopy measurements? Could they provide AFM images and root mean square (RMS) roughness of the proposed AgNWs/PI electrodes?
4. Could the authors measure the hydrophobicity of the prepared films?
5. Statistical data ("box plot" or histrogram) for a batch of devices would strengthen the manuscript.
Author Response
We thank the reviewer for the insightful comments and suggestions, which are helpful for us to further improve the quality of our manuscript. We provide the response to each comment as below.

Round 2
Reviewer 1 Report
The authors have satisfactorily revised the manuscript. I recommend this manuscript for a publication in Nanomaterials.
Reviewer 2 Report
I have no further comments.